EMBO
Molecular Medicine

# Mitochondrial glycerol 3-phosphate dehydrogenase promotes skeletal muscle regeneration

Xiufei Liu[1,†], Hua Qu[1,†], Yi Zheng[1,†], Qian Liao[1,†], Linlin Zhang[1], Xiaoyu Liao[1], Xin Xiong[1], Yuren Wang[1], Rui Zhang[1], Hui Wang[1], Qiang Tong[1], Zhenqi Liu[2], Hui Dong[3], Gangyi Yang[4] (iD), Zhiming Zhu[5], Jing Xu[1] & Hongting Zheng[1,*] (iD)

## Abstract

While adult mammalian skeletal muscle is stable due to its post-mitotic nature, muscle regeneration is still essential throughout life for maintaining functional fitness. During certain diseases, such as the modern pandemics of obesity and diabetes, the regeneration process becomes impaired, which leads to the loss of muscle function and contributes to the global burden of these diseases. However, the underlying mechanisms of the impairment are not well defined. Here, we identify mGPDH as a critical regulator of skeletal muscle regeneration. Specifically, it regulates myogenic markers and myoblast differentiation by controlling mitochondrial biogenesis *via* CaMKKβ/AMPK. mGPDH$^{-/-}$ attenuated skeletal muscle regeneration *in vitro* and *in vivo*, while mGPDH overexpression ameliorated dystrophic pathology in mdx mice. Moreover, in patients and animal models of obesity and diabetes, mGPDH expression in skeletal muscle was reduced, further suggesting a direct correlation between its abundance and muscular regeneration capability. Rescuing mGPDH expression in obese and diabetic mice led to a significant improvement in their muscle regeneration. Our study provides a potential therapeutic target for skeletal muscle regeneration impairment during obesity and diabetes.

**Keywords** diabetes; mGPDH; obesity; skeletal muscle regeneration
**Subject Categories** Metabolism; Musculoskeletal System; Regenerative Medicine

## Introduction

Adult mammalian skeletal muscle is a stable tissue that is post-mitotic; however, it actively undergoes regeneration following injury (Charge & Rudnicki, 2004). In certain diseases, such as the modern pandemics of obesity and diabetes, skeletal muscle regeneration becomes impaired, which leads to the loss of muscle function and contributes to the global burden of these diseases (Fu *et al*, 2016; Benoit *et al*, 2017). However, the mechanisms that underlie the regeneration impairment are poorly understood.

mGPDH is an integral component of the mitochondrial respiratory chain and functions as the rate-limiting step in the glycerophosphate (GP) shuttle (Eto *et al*, 1999b). Due to a different structure and cell localization, the function and regulation of this enzyme are distinct from those of cytoplasmic glycerol 3-phosphate dehydrogenase (cGPDH, also referred to as *GPD1*; Mracek *et al*, 2013). At the same time, despite the relatively simple structure of mGPDH, its functions remain largely unknown. Recently, mGPDH has been reported to be involved in hepatic glucose metabolism (Baur & Birnbaum, 2014; Madiraju *et al*, 2014). To gain a more complete understanding of mGPDH functions, we examined the role of mGPDH in skeletal muscle, which is a major insulin-sensitive tissue that plays an essential role in glucose metabolism. Our results showed that although mGPDH is vital in regulating hepatic glucose metabolism, it did not significantly affect the glucose uptake and insulin signaling within skeletal muscle (differentiated C2C12 myoblasts). Interestingly, however, the mGPDH expression significantly increased over the course of C2C12 myocyte differentiation, with an expression profile similar to that of myogenic markers (myogenin and MyHC). These differentiation-associated increases in mGPDH expression and activity were also clearly visible in mitochondrial fractions, which indicate that mGPDH might be involved in myogenic differentiation.

In the current study, we identify a novel characteristic of mGPDH in regulating myogenic differentiation and a potential therapeutic target for ameliorating muscle regeneration impairment and muscle pathology. In addition, the activation of the mGPDH/AMPK/mitochondrial biogenesis pathway of skeletal muscle might represent a new mechanism for treatment during obesity and diabetes.

1  Translational Research Key Laboratory for Diabetes, Department of Endocrinology, Xinqiao Hospital, Third Military Medical University, Chongqing, China
2  Division of Endocrinology and Metabolism, Department of Internal Medicine, University of Virginia Health System, Charlottesville, VA, USA
3  Department of Gastroenterology, Xinqiao Hospital, Third Military Medical University, Chongqing, China
4  Department of Endocrinology, The Second Affiliated Hospital, Chongqing Medical University, Chongqing, China
5  Department of Hypertension and Endocrinology, Daping Hospital, Third Military Medical University, Chongqing, China
   *Corresponding author. Tel: +8602368755709; Fax: +8602368755707; E-mail: fnf7703@hotmail.com
   †These authors contributed equally to this work

# Results

## mGPDH regulates myoblast differentiation

Our preliminary observations showed that mGPDH did not significantly influence glucose uptake or insulin signaling under both non-insulin- and insulin-treated conditions (Appendix Fig S1A and B), but its expression was augmented during the course of C2C12 myocyte differentiation (Fig 1A–D). To further explore the possibility of mGPDH involvement in myogenic differentiation, we regulated mGPDH expression by overexpression (plasmid pPR-mGPDH) or inhibition (specific siRNA si-mGPDH) in C2C12. Less cell fusion and multinuclear myotube formation events were observed in the si-mGPDH group than those in the control, and striking increases in these events accompanied overexpression (Fig 1E–G). Furthermore, the protein expression of myogenin and MyHC was also reduced by si-mGPDH during the course of differentiation (Fig 1H–K). Consistent changes in the corresponding mRNA levels following mGPDH expression perturbation were also identified (Fig 1L and M). These findings indicate that mGPDH is essential for myoblast differentiation. A major function of mGPDH is to form the GP shuttle with cGPDH. The expression of cGPDH was not significantly changed during the course of C2C12 myocyte differentiation, and the knockdown of cGPDH by siRNA did not show significant effects on C2C12 myocyte differentiation (Fig EV1A–F).

## mGPDH is essential in skeletal muscle regeneration

Myoblast differentiation occurs during muscle development and also during adulthood for muscle mass maintenance and muscle regeneration (Charge & Rudnicki, 2004). Here, we aim to identify the role of mGPDH in both stages. First, we examined the mGPDH distribution among different skeletal muscles and found that it was abundantly expressed in the gastrocnemius (GA) and quadriceps femoris (QUA), particularly in the GA (Fig EV2A); it seems that mGPDH does not match with the muscle fiber type. To further observe this issue, we costained MHC IIb (the most abundant fiber type in GA muscle) with mGPDH. The results showed the fibers were stained as three colors (Fig EV2B), which indicates that mGPDH did not match with the fiber type in GA muscle. Moreover, the expressions of MHC isoforms (MHC I, IIa, and IIb) were not significantly changed in mGPDH-depleted skeletal muscle (Fig EV2C). During muscle development, the mGPDH expression increased after birth, but only for the first few postnatal days (Fig EV2D). In the mGPDH knockout (mGPDH$^{-/-}$) mice (Fig EV2E), there were no significant differences in the body and muscle weight compared with the wild-type (WT) mice during development (Fig EV2F and G). Histological analysis also showed no differences in the muscle appearance or myofiber size between these two genotypes (Fig EV2H and I), which suggests mGPDH is not essential for muscle development.

Thus, we subsequently assessed the role of mGPDH in muscle regeneration post-injury. Both the mGPDH expression and activity were increased in GA muscle after cardiotoxin (CTX) injury and paralleled the changes of myogenic markers and developmental myosin heavy chain (Fig 2A–C), which is consistent with our observation *in vitro* (Fig 1A–D). In addition, compared with the basal expression of mGPDH in normal fibers with peripheral nuclei, the injury-induced higher expression of mGPDH was mainly localized in regenerating fibers with central nuclei (Appendix Fig S2), which indicates the injury-induced mGPDH expression predominately presented in newly formed myofibers. Although both the mGPDH$^{-/-}$ and WT mice exhibited extensive muscle damage at day 3 post-injury, the mGPDH$^{-/-}$ mice showed a delay in the disappearance of necrotic fibers and inflammatory cells and had fewer and more unevenly distributed newly formed myofibers with multiple centrally located nuclei at day 7 (Fig 2D–F). The immunofluorescence of desmin, an intermediate filament protein in newly generated myofibers (Liu *et al*, 2012), further confirmed the impaired muscle regeneration in mGPDH$^{-/-}$ mice (Fig 2G). At day 14, the muscle weight was decreased (Fig 2H), while the collagen deposition was increased (Fig 2I) in the mGPDH$^{-/-}$ mice. These results suggested that mGPDH loss attenuates muscle regeneration. At the same time, expressions of the satellite cell marker paired box protein 7 (PAX7; Zhang *et al*, 2016; Bi *et al*, 2017) and the satellite cell activation marker myoblast determination protein (MyoD; Zhang *et al*, 2016; Bi *et al*, 2017) were not different between the mGPDH$^{-/-}$ and WT mice (Appendix Fig S3A–F), which suggests that mGPDH has no significant effects on myoblast quantity and activation. However, the differentiation markers myogenin and myh3 (Park *et al*, 2016) were reduced in the mGPDH$^{-/-}$ mice (Fig 2J and K), which is consistent with our *in vitro* data and indicates that mGPDH deletion inhibits skeletal muscle regeneration by diminishing myoblast differentiation.

Next, we activated mGPDH *via* AAV in mdx mice, which represent a model of Duchenne muscular dystrophy, in which there is a persistent damage and loss of myofibers induced by the *Dmd* gene mutation (Barton *et al*, 2002; Duddy *et al*, 2015; Novak *et al*, 2017). The basal expression levels of mGPDH and myogenin were increased in the mdx mice compared to the normal mice, which indicated an activated regeneration process that was insufficient to compensate (Appendix Fig S4A and B). The overexpression of mGPDH *via* intramuscular injection of AAV into the GA muscle induced a further increase in myogenin and myh3 expression (Fig 2L); in line with this finding, the number of small regenerating fibers and the variability in the myofiber size were decreased (Fig 2M), and the distribution of the cross-sectional area (CSA) shifted to the right (Fig 2N). Moreover, the mRNA and protein levels of utrophin, an indicator of regeneration in mdx mice (Durko *et al*, 2010), were increased (Fig 2O and P), while muscle fibrosis decreased (Fig 2Q). Furthermore, systematically up-regulating mGPDH *via* tail vein injection of AAV improved the exercise capacity of the mdx mice (Appendix Fig S5 and Fig 2R). Taken together, these *in vivo* data of mGPDH deletion and overexpression suggest that mGPDH plays a pivotal role in regulating myoblast differentiation and muscle regeneration.

## mGPDH effects occur *via* the CaMKKβ/AMPK control of mitochondrial biogenesis

To gain further insights into the underlying molecular mechanisms, we subsequently assessed a number of the common factors related to myoblast differentiation, such as the cell cycle, apoptosis, autophagy, insulin-like growth factor-1 (IGF-1), and mitochondrial biogenesis (Musaro *et al*, 2001; Kim *et al*, 2010; Hochreiter-Hufford *et al*, 2013; Zhang *et al*, 2014; Garcia-Prat *et al*, 2016). mGPDH had no significant effects on the cell cycle, apoptosis, autophagy, and

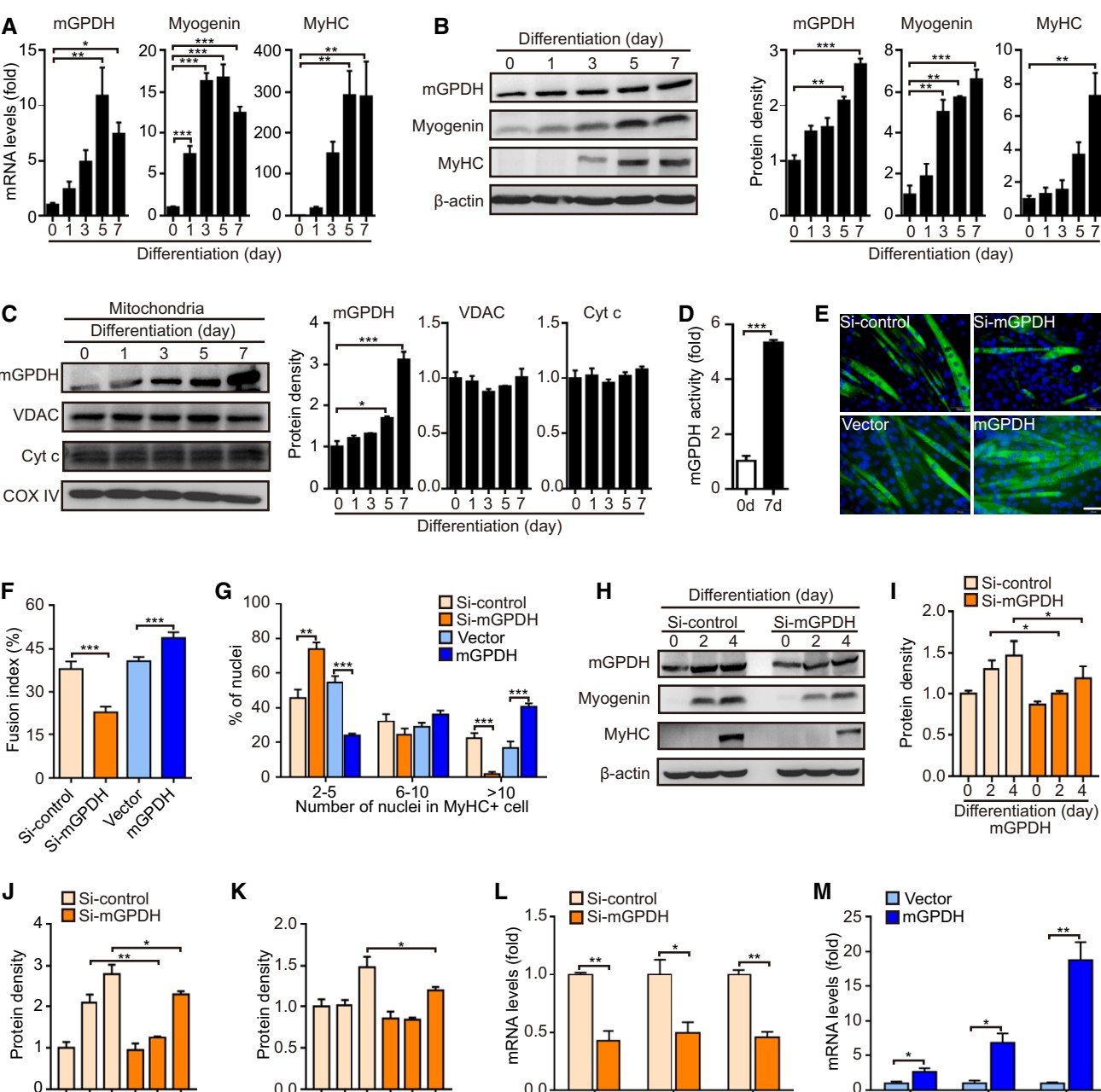

**Figure 1. mGPDH regulates myoblast differentiation.**

A, B   qRT–PCR (A) and immunoblot (B) of mGPDH, myogenin, and myosin heavy chain (MyHC) levels during C2C12 myocyte differentiation. Quantification represents the levels of the indicated protein normalized to β-actin.

C      Immunoblot of mGPDH, voltage-dependent anion channel (VDAC), and cytochrome c (Cyt C) levels in mitochondrial lysate during C2C12 myocyte differentiation. Quantification represents the levels of the indicated protein normalized to COX IV.

D      Activity assay of mGPDH at days 0 and 7 after C2C12 myocyte differentiation.

E–G    Representative images of MyHC immunofluorescence (E) of C2C12 myocyte transfected with the siRNA or the overexpression plasmid for mGPDH; the fusion index (F) and the distribution of nuclei per myotube (G) were calculated at day 5 after differentiation.

H–K    Immunoblot of mGPDH, myogenin, and MyHC in C2C12 myocytes transfected with siRNA targeting mGPDH. Quantification (I–K) represents the levels of the indicated protein normalized to β-actin at the indicated day after differentiation.

L, M   qRT–PCR analysis of mGPDH, myogenin, and MyHC in C2C12 myocytes transfected with the siRNA or the overexpression plasmid for mGPDH at day 4 after differentiation.

Data information: Data are presented as the mean ± s.e.m. Scale bars represent 50 μm in panel (E). In panels (A–D) and (H–M), $n = 3$; in panels (E–G), $n = 15$. *$P < 0.05$, **$P < 0.01$, ***$P < 0.001$. Unpaired $t$-test was used for all analyses except in panel (G), where Kolmogorov–Smirnov test was used.

Source data are available online for this figure.

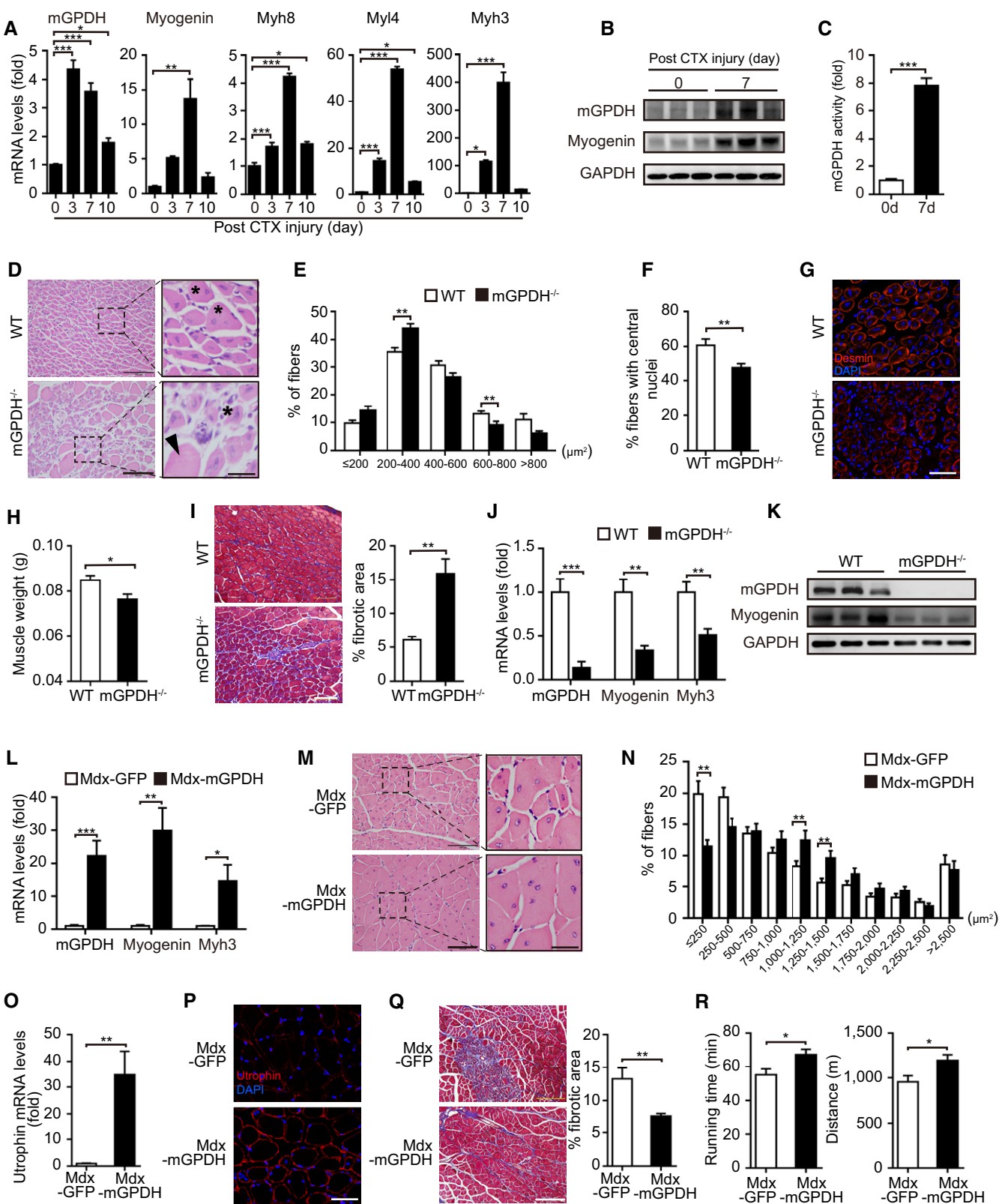

**Figure 2.**

IGF-1 receptor expression (Appendix Fig S6A–D), but obviously changed the mitochondrial content of C2C12 myocytes (Fig 3A). Moreover, it regulated the expression of nuclear-encoded oxidative phosphorylation (OXPHOS) subunits (*NDUFS8*, *SDHb*, *Uqcrc1*, *COX5*, and *ATP5a*; Fig 3B), despite no substantial impact on the mitochondrial genomes (*ND1*, *Cytb*, *COX1*, and *ATP6*; Appendix Fig

◄

**Figure 2.  mGPDH is essential to skeletal muscle regeneration.**

A, B    qRT–PCR (A) and immunoblot (B) of mGPDH, myogenin, and developmental myosin heavy chain (myh8, myl4, and myh3) in gastrocnemius (GA) muscle from C57BL/6J mice at the indicated day after CTX intramuscular injection.

C       Activity assay of mGPDH in GA muscle from C57BL/6J mice at days 0 and 7 after CTX injection.

D–G     Representative images of the H&E staining (arrowhead, necrotic myofibers; asterisks, regenerating fibers) (D), distribution of the fiber cross-sectional area (CSA) (E), percentage of myofibers with central nuclei (F), and immunofluorescence staining of desmin (green) (G) in GA muscle from WT and mGPDH$^{-/-}$ mice at day 7 post-CTX injection.

H, I    Muscle weight (H) and trichrome staining (I) in GA muscle from WT and mGPDH$^{-/-}$ mice at day 14 post-CTX injection. Quantification represents the fibrotic areas.

J, K    qRT–PCR (J) and immunoblot (K) for mGPDH, myogenin, and myh3 in GA muscle from WT and mGPDH$^{-/-}$ mice at day 7 post-CTX injection.

L–Q     qRT–PCR for mGPDH, myogenin, and myh3 (L), H&E staining (M), distribution of the fibers CSA (N), qRT–PCR (O), and immunofluorescence staining (P) for utrophin and trichrome staining (Q) in GA muscle from mdx mice 4 weeks after AAV-mGPDH intramuscular injection.

R       Exercise capacity of mdx mice 6 weeks after AAV-mGPDH tail vein injection.

Data information: Data are presented as the mean ± s.e.m. Scale bars represent 100 μm (25 μm for magnification insets) in panels (D, I, M, and Q) and 50 μm in panels (G, P). In panels (A–C), *n* = 3; in panels (D–R), *n* = 6 mice per group; in panels (D–F, M, and N), three sections were obtained per mouse. *$P < 0.05$, **$P < 0.01$, ***$P < 0.001$. Unpaired *t*-test was used for all analyses except in panels (E, N), where the Kolmogorov–Smirnov test was used.

Source data are available online for this figure.

S7A and B). The mitochondrial respiration rate further confirmed these links to mitochondrial biogenesis (Fig 3C). AMP-activated protein kinase (AMPK) is a key regulator of nuclear-encoded OXPHOS subunits and mitochondrial function (Xiao *et al*, 2011; Lin *et al*, 2012; Gomes *et al*, 2013; Mottillo *et al*, 2016), and it has also been reported to influence myoblast differentiation (Mounier *et al*, 2013). Therefore, we assessed whether AMPK is involved in mGPDH effects. As shown in Fig 3D, mGPDH expression significantly activated AMPK and its downstream acetyl-CoA carboxylase (ACC), as well as the mitochondrial biogenesis marker peroxisome proliferator-activated receptor-γ coactivator-1α (PGC1α; Fig 3D–F). Strikingly, the activated mitochondrial biogenesis caused by mGPDH overexpression, including increased PGC1α, mitochondrial content, and nuclear-encoded OXPHOS, was abrogated when the AMPK inhibitor compound C was applied (Fig 3G–I), which indicates that the effects of mGPDH on nuclear-encoded OXPHOS subunits and mitochondrial biogenesis are AMPK-dependent. Previous studies have reported that to a large extent, AMPK regulated mitochondrial biogenesis mainly through the modulation of PGC1α activity by the NAD$^+$/NADH ratio (Iwabu *et al*, 2010; Meng *et al*, 2013; Woldt *et al*, 2013). Our results showed that mGPDH loss- and

gain-of-function manipulations affected the NAD$^+$/NADH ratio (Fig 3J). Moreover, PGC1α acetylation was altered when mGPDH expression changed (Fig 3K). Mitochondrial biogenesis is critical for skeletal muscle differentiation. It activates myoblast differentiation markers by suppressing c-myc expression, which represses myoblast differentiation through direct binding to the promoters or enhancers of myogenin (Miner & Wold, 1991; Seyer *et al*, 2010; Ravel-Chapuis *et al*, 2014). In our observation, mGPDH overexpression repressed c-myc expression and increased myogenin expression and myoblast differentiation, and these effects were abolished by the AMPK inhibitor compound C (Fig 3L–P).

AMPK activity is dependent on the phosphorylation of AMPKα (Thr172) by AMPK kinases (AMPKKs; Iwabu *et al*, 2010). To further clarify the link between mGPDH and AMPK, two major AMPKKs, liver kinase B1 (LKB1) and Ca2$^+$/calmodulin-dependent protein kinase kinase (CaMKK) β (Hawley *et al*, 2005; Kahn *et al*, 2005; Woods *et al*, 2005), were assessed. LKB1 siRNA did not hinder the regulation of mGPDH on AMPK (Appendix Fig S8), in contrast to the CaMKKβ-specific inhibitor STO-609 (Fig 3Q). We subsequently explored how mGPDH regulates cytoplasmic CaMKKβ. Previous studies have demonstrated that intracellular Ca$^{2+}$ plays an

►

**Figure 3.   mGPDH effect occurs via the CaMKKβ/AMPK control of mitochondrial biogenesis.**

A–F     Mitochondrial DNA (A), nuclear-encoded OXPHOS genes (B), respirometry analysis (C), and immunoblots of mGPDH, phospho-Thr172 AMPK (p-AMPK), total AMPK (AMPK), phospho-Ser79-ACC (p-ACC), total ACC and PGC1α, and corresponding quantifications represent mGPDH, p-AMPK, p-ACC, and PGC1α protein levels (D–F) in C2C12 myocytes transfected with siRNA or plasmid for mGPDH 24 h after differentiation.

G–I     Immunoblots of p-AMPK, p-ACC, and PGC1α and corresponding quantifications represent p-AMPK, p-ACC, and PGC1α protein levels (G), mitochondrial DNA (H), and nuclear-encoded OXPHOS genes combined by *NDUFS8*, *SDHb*, *Uqcrc1*, *COX5b*, and *ATP5a1* (I) in C2C12 myocytes transfected by mGPDH plasmid with the AMPK inhibitor compound C (CC) 24 h after differentiation.

J, K    NAD$^+$/NADH ratio (J) and immunoprecipitation analysis for PGC1α acetyl-lysine (Ac-Lys) level (K) in C2C12 myocytes transfected with siRNA or plasmid for mGPDH 24 h after differentiation.

L–P     Immunoblot of c-myc and myogenin (L) and corresponding quantifications represent c-myc and myogenin protein levels (M), representative images of MyHC immunofluorescence (N), fusion index (O), and the distribution of nuclei per myotube (P) in C2C12 myocytes transfected with mGPDH plasmid with the AMPK inhibitor CC at 24 h (L, M) or 72 h (N–P) after differentiation.

Q       Immunoblots of p-AMPK, p-ACC, PGC1α, and myogenin in C2C12 myocytes transfected with mGPDH plasmid with the CaMKKβ inhibitor STO-609 at 24 h after differentiation. Quantifications represent p-AMPK, p-ACC, PGC1α, and myogenin protein levels.

R       Immunoblots of p-AMPK and p-ACC in C2C12 myocytes transfected with mGPDH plasmid with the Ca$^{2+}$ chelator BAPTA-AM at 24 h after differentiation. Quantifications represent p-AMPK and p-ACC protein levels.

Data information: Data are presented as the mean ± s.e.m. Scale bars represent 50 μm in panel (N). In panels (A, B, D–M, Q, and R), *n* = 3; in panel (C), *n* = 10; in panels (N–P), *n* = 15. *$P < 0.05$, **$P < 0.01$, ***$P < 0.001$, n.s.: not significant. Unpaired *t*-test was used in panels (A–C, E, F, and J); one-way ANOVA with Tukey's comparison test was used in panels (G–I, M, O, Q, and R); and the Kolmogorov–Smirnov test was used in panel (P).

Source data are available online for this figure.

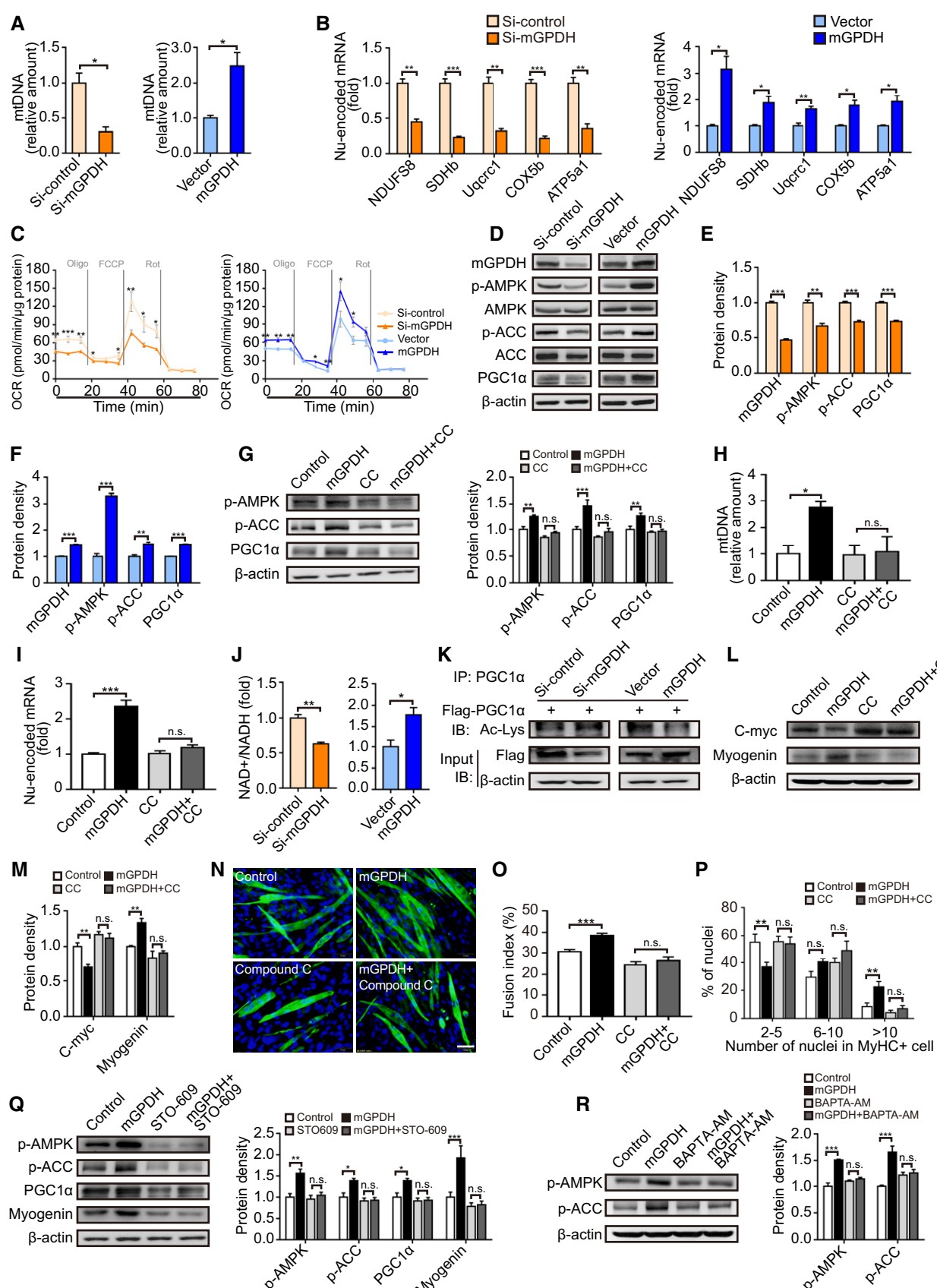

**Figure 3.**

important role in the crosstalk between mitochondria and the cytoplasm (Ganitkevich, 2003; Fieni et al, 2014; Stefani et al, 2016), and $Ca^{2+}$ has also been reported to serve as the initiating factor during the cascade involving CaMKKβ-activated AMPK (Iwabu et al, 2010; Marcelo et al, 2016). Therefore, we used a cell membrane-permeable $Ca^{2+}$ chelator, BAPTA-AM, to eliminate intracellular free $Ca^{2+}$ and found that mGPDH-induced AMPK activation was abolished (Fig 3R). Together, these data suggest that the regulation of myoblast differentiation by mGPDH occurs via CaMKKβ/AMPK control of mitochondrial biogenesis.

### Rescuing mGPDH deficiency improves skeletal muscle regeneration during obesity and diabetes

Based on the observed effects of mGPDH on myoblast differentiation and muscle regeneration, we then explored its role under pathological conditions. Skeletal muscle regeneration was impaired in obese and diabetic mice (Fig EV3A–H), which was consistent with previous observations (Aragno et al, 2004; Fu et al, 2016). In patients, approximately 60% of obese patients have a low muscle mass, decreased muscle strength, and poor physical function (Chiu et al, 2016; Fu et al, 2016; Lee et al, 2016; Benoit et al, 2017). Our results showed a lower mGPDH expression in the GA muscle of obese patients compared with healthy subjects (ChiCTR-ROC-17010719; Fig 4A and B, and Appendix Table S1). These results were replicated in obese (ob/ob and HFD), diabetic (STZ), and diabetic–obese (db/db) mice, accompanied by myogenin reduction (Fig 4C). mGPDH activation during the regeneration process was also defective in these models (Fig 4D). In contrast, the cGPDH expressions were not notably altered in obese (HFD), diabetic (STZ), and mdx mice. Moreover, the expressions did not change during the regeneration process post-CTX injection (Appendix Fig S9). Thus, we observed the effects of rescuing mGPDH expression on muscle regeneration during obesity and diabetes. By day 7 post-CTX injury, the AAV-mGPDH-treated group had higher expressions of myogenin and myh3 in the GA muscle from both the obese (HFD and ob/ob) and diabetic (STZ) mice (Figs 4E and I, and EV4A). Histological analysis showed enhanced muscle regeneration, evidenced by a reduction of necrotic fibers, a right-shifted CSA distribution, and an increased number of newly formed myofibers with central myonuclei across these animal models (Figs 4F–H and J–L, and EV4B–D). The AAV-mGPDH mice also exhibited a higher muscle weight

compared to the controls at day 14 (Fig 4M), which further confirmed that they underwent a more efficient regeneration. Moreover, the AMPK pathway responses downstream of mGPDH were also activated by AAV-mGPDH (Fig 4N), corroborating our in vitro mechanistic findings. In addition, previous studies have shown that inflammatory signaling is an important player in the regulation of tissue repair and regeneration (De Bleecker & Engel, 1994). However, although mGPDH ablation (mGPDH$^{-/-}$ mice) or overexpression (AAV-mGPDH) showed relatively decreased or increased trends of cytokine genes, respectively, their expressions were not statistically significant with the exception of IL-1β in the mGPDH$^{-/-}$ mice (Fig EV5A–C).

## Discussion

Both obese and diabetic patients have been proven to suffer from delayed skeletal muscle regeneration, which led to impaired muscle function and a poor prognosis (Hu et al, 2010; Fu et al, 2016). The present study identified mGPDH as a pivotal regulator of myoblast differentiation that contributes to the process of skeletal muscle regeneration. The mechanical studies indicated that the effects were mainly through CaMKKβ/AMPK-controlled mitochondrial biogenesis. Importantly, the mGPDH expression of skeletal muscle was reduced in patients and animal models of obesity and diabetes, and rescuing mGPDH expression led to a significant improvement in muscle regeneration.

Our findings suggest that mGPDH is a critical regulator of skeletal muscle regeneration through the regulation of CaMKKβ/AMPK-controlled mitochondrial biogenesis. mGPDH deletion attenuated skeletal muscle regeneration in vitro and in vivo, while its overexpression ameliorated dystrophic pathology in mdx mice. Interestingly, our study showed an increased basal expression of myogenin in mdx mouse muscles compared with normal mice, which is consistent with previous observations (Turk et al, 2005). This result indicates an activated regeneration process; however, it remains insufficient to compensate due to persistent damage induced by the Dmd gene mutation in mdx mice (Barton et al, 2002; Duddy et al, 2015; Novak et al, 2017). Therefore, as shown in our study, the basal level of mGPDH is also increased in mdx mice. The regulatory role of mitochondrial biogenesis on muscle regeneration has been proven by several studies (Cerletti et al, 2012; Stein & Imai, 2014;

---

**Figure 4.  Rescuing mGPDH deficiency improves skeletal muscle regeneration during obesity and diabetes.**

A–C   Immunoblot (A, C) and IHC (B) of mGPDH and myogenin in GA muscles from obese patients (A, B) and the indicated mice (C).

D       qRT–PCR of mGPDH in GA muscle of the indicated mice at days 0 and 3 after CTX intramuscular injection.

E–H   Experimental setup (E, upper panel); qRT–PCR of mGPDH, myogenin, and myh3 (E, bottom panel); H&E staining (arrowhead, necrotic myofibers; asterisks, regenerating fibers) (F); distribution of the fiber CSA (G); and percentage of myofibers with central nuclei (H) in GA muscle from AAV-mGPDH-treated HFD-fed mice at day 7 after CTX intramuscular injection.

I–M   Experimental setup (I and M, upper panels); qRT–PCR of mGPDH, myogenin, and myh3 (I, bottom panel); H&E staining (arrowhead, necrotic myofibers; asterisks, regenerating fibers) (J); distribution of the fibers CSA (K); percentage of myofibers with central nuclei (L); and muscle weight (M, bottom panel) in GA muscle from AAV-mGPDH-treated STZ-treated mice at days 7 (I–L) and 14 (M) after CTX intramuscular injection.

N       Immunoblots of mGPDH, p-AMPK, p-ACC, PGC1α, and myogenin for the experiment described in (E).

Data information: Data are presented as the mean ± s.e.m. Scale bars represent 200 μm in panel (B) and 100 μm (25 μm for magnification insets) in panels (F, J). In panels (A, B), obese patients (n = 11) and normal subjects (n = 18); in panels (C, D), n = 3 mice per group; in panels (E–L and N), n = 6 mice per group; in panel (M), n = 4 mice per group; in panels (B, F–H, and J–L), three sections were obtained per mouse. *P < 0.05, **P < 0.01, ***P < 0.001. Unpaired t-test was used in panels (D, E, H, I, L, and M); the Wilcoxon test was used in panel (B); and the Kolmogorov–Smirnov test was used in panels (G, K).

Source data are available online for this figure.

Zhang et al, 2014; Katajisto et al, 2015). AMPK, as the most important regulator of mitochondrial function, modulates muscle regeneration via mitochondrial biogenesis as previously reported (Woldt et al, 2013). Our results demonstrated that mGPDH regulates

skeletal muscle regeneration through AMPK-controlled mitochondrial biogenesis, which may be a new upstream regulation mechanism of AMPK on muscle regeneration. Our study also suggests that the effect of mGPDH is mainly through CaMKKβ-triggered AMPK

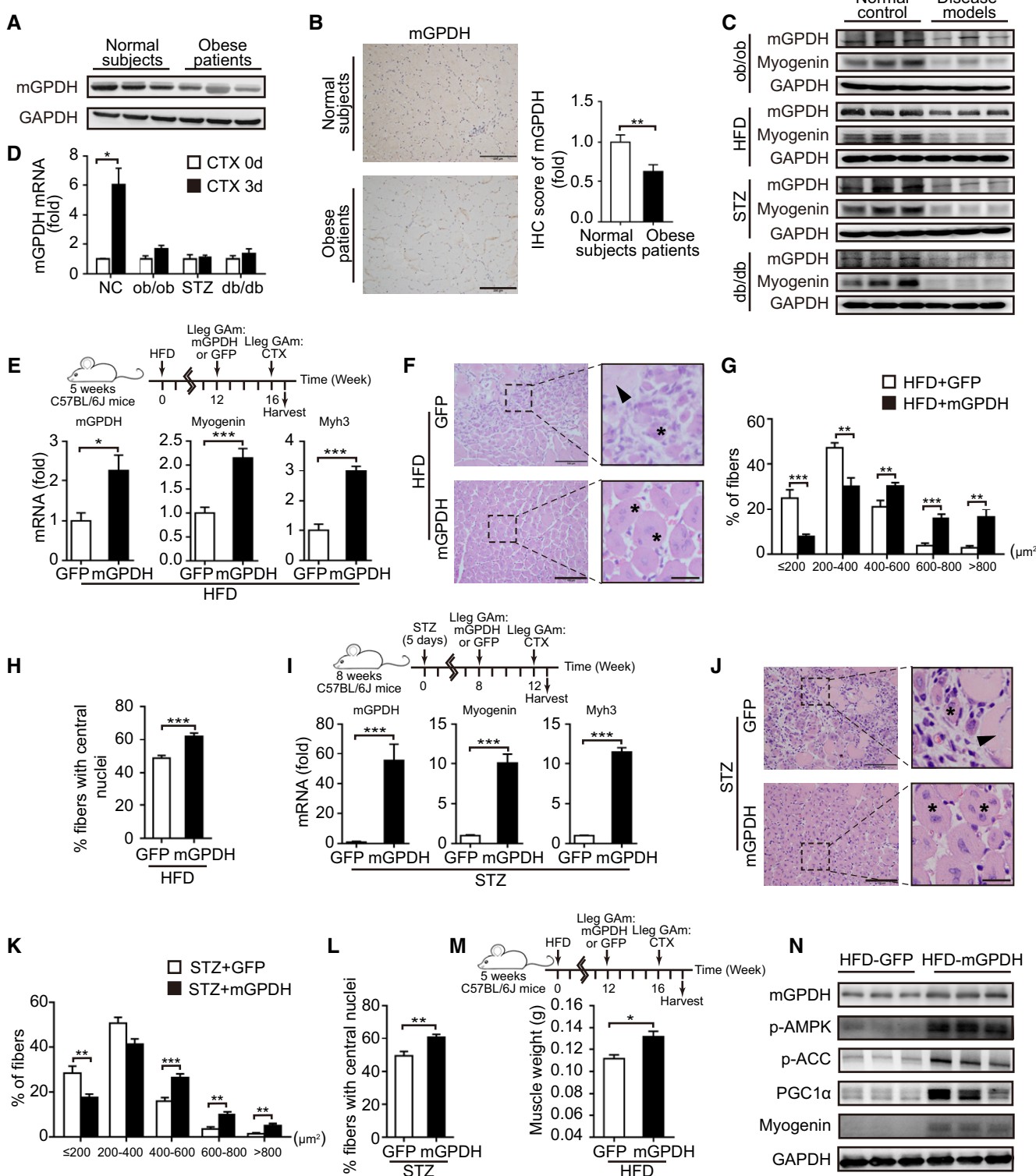

**Figure 4.**

activation, while LKB1 does not seem to participate in this process. As a calmodulin-dependent kinase, CaMKKβ activation depends on the intracellular $Ca^{2+}$ concentration (Marcelo et al, 2016). Previous studies have reported that mGPDH regulates the intracellular $Ca^{2+}$ concentration in pancreatic beta cells (Eto et al, 1999a,b), which further confirmed our observation regarding the regulation of mGPDH on CaMKKβ. Notably, although CaMKKβ and LKB1 are known to be two major AMPKKs (Hawley et al, 2005; Kahn et al, 2005; Woods et al, 2005), the participation of other AMPKKs, such as AMP, PI3K, and PKC, cannot be completely ruled out (Zou et al, 2003; Nishino et al, 2004; Oakhill et al, 2011). Our study strengthened previous observations regarding the impact of mitochondrial biogenesis on myoblast differentiation and muscle regeneration. An enhanced mitochondrial mass and/or function promoted myoblast differentiation and muscle regeneration, while attenuated mitochondrial biogenesis inhibited them by manipulating the myogenic regulatory factors. Our results are consistent with these findings and further confirmed an important role of mitochondrial biogenesis on functional muscle regeneration (Cerletti et al, 2012; Duguez et al, 2012; Varaljai et al, 2015).

Our study provides a potential therapeutic target for skeletal muscle regeneration impairment during obesity and diabetes. Deficient mGPDH expression was identified in the skeletal muscles of patients and animal models of obesity and diabetes. Rescuing mGPDH expression substantially promotes muscle regeneration in HFD, ob/ob, and STZ mice, which indicates the therapeutic potential of targeting mGPDH. Previous studies have suggested that several common treatments for obesity and diabetes, such as moderate exercise, metformin, and GLP-1 agonists, are capable of promoting muscle regeneration by controlling mitochondrial biogenesis (Tong et al, 2011; Wu et al, 2011; Yamamoto et al, 2013). Therefore, further investigations are required to explore the potential relationships between mGPDH and the effects of these therapies on muscle regeneration. Moreover, the application of mGPDH activators for these diseases should be investigated in the future.

Collectively, our study revealed a novel functional role of mGPDH in regulating myogenic differentiation and skeletal muscle regeneration. In addition, activation of the mGPDH/AMPK/mitochondrial biogenesis pathway in skeletal muscle might be a mechanism of importance for ameliorating muscular frailty during obesity and diabetes based on our in vitro explorations and animal model recapitulations. Direct therapeutic targeting of mGPDH may also have therapeutic potential.

# Materials and Methods

## Cell culture

C2C12 myocytes were purchased from the Cell Bank of the Chinese Academy of Sciences and were maintained at subconfluent densities in growth medium consisting of Dulbecco's modified Eagle's medium (DMEM; Gibco) supplemented with 10% fetal bovine serum (FBS; Gibco) in a 5% $CO_2$ incubator at 37°C. To induce myogenic differentiation, cells were grown to 95% confluence in growth medium and then cultured in differentiation medium composed of DMEM and 2% horse serum (Gibco). The differentiation medium was changed every 48 h. All cell identities were confirmed and cultured as recommended by the supplier. Mycoplasma determination was performed by Shanghai Biowing Applied Biotechnology Co., and no mycoplasma contamination was identified in these cells.

## Cell transfection and treatment

For the knockdown, the cells were seeded in 6-well culture plates and transiently transfected with 50 ng per well of siRNA oligonucleotides with RNAiMAX (Life Technology) according to the manufacturer's instructions. mGPDH-specific, LKB1-specific, and corresponding negative-control siRNAs were synthesized by Qiagen, and the sequences are presented in Appendix Table S2. For overexpression, cells were seeded in 6-well culture plates and transfected with 1 μg of plasmid using Lipofectamine 3000 reagent (Invitrogen) according to the manufacturer's instructions. Plasmids of the pRP-mGPDH and control vector were generated by VectorBuilder. Compound C and STO-609 were purchased from Sigma-Aldrich. BAPTA-AM was purchased from TCI.

## Glucose uptake

Cells were serum-starved for 3 h in culture medium (no glucose) and subsequently incubated with 100 μM of fluorescent deoxyglucose 2-NBDG (Invitrogen) with or without 100 nM of insulin (Biosharp) for 1 h. The cells were washed and harvested in ice-cold PBS. The fluorescence intensity was determined using a flow cytometer (MoFlo XDP; Beckman Coulter) at excitation and emission wavelengths of 485 and 538 nm, respectively.

## Cell mitochondrial protein isolation

The mitochondria of cells were isolated using a Mitochondria Isolation Kit (Beyotime) according to the manufacturer's recommendation. Briefly, the cells were collected, re-suspended, and homogenized in 1 ml of mitochondrial separation reagent supplemented with PMSF, and the suspension was then centrifuged at 600 g for 10 min at 4°C to remove nuclei and cell debris. The supernatant was re-centrifuged at 11,000 g for 10 min at 4°C, and the remaining pellet contained the mitochondria. Mitochondrial lysis solution (50 μl) was added to break down the mitochondrial proteins, and protein concentrations were detected with the BCA Protein Assay Kit (Beyotime).

## Cell cycle and apoptosis analyses

For the cell cycle analysis, the cells were detached by trypsin and fixed in 70% ethanol overnight. The cells were washed and re-suspended in 0.5 ml of PBS that contained 100 μg/ml of RNase A and 5 μg/ml of propidium iodide (PI) for 30 min. The DNA contents were measured by flow cytometry. Apoptosis was detected using the FITC Annexin V Apoptosis Detection Kit (Becton Dickinson) according to the manufacturer's instructions. Briefly, cells were harvested and washed twice with ice-cold PBS, re-suspended in 100 μl of binding buffer, and incubated with 5 μl of Annexin V–FITC and 5 μl of PI. After a 30-min incubation at room temperature in the dark, 400 μl of binding buffer was added to each tube, and the samples were immediately analyzed via flow cytometry.

## Cell oxygen consumption rate (OCR) measurement

The cellular oxidation state was measured using a Seahorse XF96 extracellular flux analyzer (Seahorse Biosciences) according to the manufacturer's protocol. In brief, 10,000 cells were seeded in each well of XF 96-well microplates (Seahorse Bioscience). The final concentrations of the mitochondrial inhibitors were 1 μM of oligomycin, 3 μM of FCCP, and 0.5 of μM rotenone. Basal respiration indicates the baseline oxygen consumption reading prior to compound injection. Maximal respiration represents the maximum OCR measurement value after FCCP injection. After detection, the cell protein concentrations were assessed and the OCR was adjusted accordingly.

## Mitochondrial content

Total DNA was extracted using a DNA purification kit (Promega) according to the manufacturer's instructions. The mitochondrial DNA copy number was determined by real-time PCR using primers specific for the mitochondrial cytochrome c oxidase subunit 2 (COX2) gene as previously described (Price et al, 2012) and was normalized to the nuclear copy number using primers specific for the ribosomal protein s18 (S18) nuclear gene. A complete list of the primer sequences is presented in Appendix Table S2.

## NAD$^+$/NADH measurements and PGC1α acetylation assay

The NAD$^+$ and NADH contents were measured using a NAD$^+$/NADH Assay Kit (Beyotime) according to the manufacturer's recommendation. The amounts of PGC1α acetylation were performed as previously described (Woldt et al, 2013). Briefly, cells were transfected with plasmid of Flag-PGC1α expression vector, and where indicated, mGPDH siRNA or plasmid was used to knock down or overexpress mGPDH as indicated. The cells were harvested, and the PGC1α acetylation was determined by immunoprecipitation of lysates with anti-PGC1α antibody (Novus NBP1-04676, 2 μg per sample), followed by Western blot analysis using antibodies to acetylated lysine (1:500, Santa Cruz, sc-32268). The input was blotted with antibody to Flag-epitope tag (1:1,000, Cell Signaling Technology, 8146s).

## Mice

All mouse experiments were performed in accordance with protocols approved by the Laboratory Animal Welfare and Ethics Committee of the Third Military Medical University. mGPDH$^{-/-}$, ob/ob, and mdx mice were purchased from the Model Animal Research Center of Nanjing University, db/db mice were a gift from Professor Gangyi Yang (Chongqing Medical University), and C57BL/6J mice were purchased from Beijing HFK Bioscience Co. Male mice were used in all experiments with the exception of mGPDH$^{-/-}$ and their littermate controls, where male and female mice were equally used. All mice were housed with a 12-h dark/light cycle with food and water ad libitum and were randomly allocated to the indicated groups. Blinding was used for the analysis of all animal experiments with the exception of the qRT–PCR and immunoblot analyses.

## HFD and STZ mouse models

C57BL/6J mice were placed on a high-fat diet [HFD, 60 kcal% from fat, purchased from Research Diets (New Brunswick)] at the age of 5 weeks and were maintained on the same diet for 12 weeks. Diabetes was induced via the intraperitoneal (i.p.) injection of streptozotocin (STZ, 50 mg/kg, pH 4.5, dissolved in sodium citrate; Sigma) for five consecutive days. Two weeks after STZ injection, the fasting plasma glucose (FPG, 8 h fast) was measured, and the mice with an FPG level greater than 250 mg/dl (13.9 mmol/l) were considered diabetic (Zheng et al, 2011).

## Skeletal muscle injury

Animals were anesthetized using pentobarbital sodium (50 mg/kg) via i.p. injection. To induce muscle injury, 50 μl of 10 μM cardiotoxin (CTX; Sigma) was injected using an insulin syringe (U-100; Becton Dickinson) into muscles (Rozo et al, 2016).

## AAV gene transfer

To avoid an in vivo immune response, adeno-associated virus (AAV)-9, which is specific to skeletal muscle with broad myofiber-specific tropism, was used (Kotterman & Schaffer, 2014; Riaz et al, 2015). The recombinant mGPDH-encoding AAV9 and the control vectors (AAV9-GFP) were generated by GENE Company. After anesthetization, AAV particles ($8 \times 10^{10}$ v.g.) were singly injected into the gastrocnemius (GA) muscles. For tail vein injection, the volume of AAV9 was $1.2 \times 10^{11}$ v.g.

## Treadmill test

The treadmill test was conducted using the ZH-PT animal running experiment (Zhenghua Biologic Apparatus Facilities) at 15 degrees downhill. Mice were warmed up at 5 m/min for 2 min, and then ran on the treadmill at 7 m/min for 2 min, 8 m/min for 2 min, and 12 m/min for 5 min. The speed was subsequently increased by 1 m/min to a final speed of 20 m/min. Exhaustion was defined as the inability of the animal to remain on the treadmill despite electrical prodding for 10 s. The running time and distance were calculated.

## H&E, Masson's trichrome, and immunohistochemical analyses

Hematoxylin and eosin (H&E), Masson's trichrome, and immuno-histochemical (IHC) staining were performed as previously described (Zheng et al, 2011). Regenerating myofibers were defined as myofibers that contained central nuclei, and the cross-sectional area (CSA) of the regenerating myofibers was determined using ImageJ (NIH) software. Nuclei per fiber were counted as the average number of central nuclei in each regenerating myofiber.

## Western blot and antibodies

For protein extraction, the cells were lysed in sample buffer [50 mM of Tris–HCl (pH 6.8), 2% SDS, 10% glycerol, 100 mM of dithiothreitol, and 0.1% bromophenol blue], tissue lysates were prepared as previously described (Wang et al, 2016), and protein concentrations

**The paper explained**

**Problem**
While adult mammalian skeletal muscle is stable due to its post-mitotic nature, muscle regeneration serves a critical role in maintaining functional fitness throughout life. During certain diseases, such as the modern pandemics of obesity and diabetes, the regeneration process becomes impaired, which leads to the loss of muscle function and contributes to the global burden of these diseases. However, the underlying mechanisms remain poorly defined.

**Results**
In this study, we report that mitochondrial glycerol 3-phosphate dehydrogenase (mGPDH, also referred to as *GPD2*) is a critical regulator for skeletal muscle regeneration. Specifically, mGPDH promotes myogenic gene expression and myoblast differentiation by controlling mitochondrial biogenesis via CaMKKβ/AMPK. mGPDH deletion attenuated skeletal muscle regeneration *in vitro* and *in vivo*, while activating mGPDH ameliorated dystrophic pathology in mdx mice. Moreover, skeletal muscle mGPDH expression was reduced in patients and animal models of obesity and diabetes, and its restoration resulted in improved muscle regeneration.

**Impact**
Our study indicates a novel characteristic of mGPDH in regulating myogenic differentiation and identifies a potential complementary strategy for enhancing muscle regeneration and ameliorating muscle pathology. In addition, activation of the mGPDH/AMPK/mitochondrial biogenesis pathway of skeletal muscle may represent a new mechanism for treatment during obesity and diabetes.

were measured using the BCA Protein Assay Kit (Beyotime). Extracted protein lysates were resolved by SDS–PAGE and immunoblotted with the indicated primary antibodies and their corresponding HRP-conjugated secondary antibodies. Blots were developed with chemiluminescent HRP substrate (Millipore) and imaged using a Fusion FX5s system (Vilber Lourmat). The following antibodies were used: mGPDH (1:1,000, sc-390830), Myogenin (1:1,000, sc-12732), cGPDH (1:500, sc-376219), IGF-1R (1:1,000, sc-81464), GAPDH (1:10,000, sc-20357), β-actin (1:2,000, sc-47778), and c-myc (1:1,000, sc-42) from Santa Cruz Biotechnology; MyHC (1:2,000, M4276) from Sigma; Akt (1:1,000, #4691), p-Akt (Thr308, 1:1,000, #13038), IRS1 (1:1,000, #2382), p-IRS1 (Ser307, 1:1,000, #2381), VDAC (1:1,000, #4866), Cyt c (1:1,000, #4280), AMPKα (1:2,000, #2532), p-AMPKα (Thr172, 1:2,000, #2535), ACC (1:1,000, #3676), p-ACC (Ser79, 1:1,000, #11818), Flag (1:1,000, #8146), and LC3B (1:1,000, #2775) from Cell Signaling Technology; PGC1α (1:1,000, ab54481) and mGPDH (1:10,000, ab188585) from Abcam; and COX IV (1:500, AC610) from Beyotime. The relative band intensities were quantified using the Fusion FX5s system (Vilber Lourmat).

**Quantitative real-time PCR (qRT–PCR)**

Total RNA was isolated from cells or tissues using RNAiso Plus (Takara), according to the manufacturer's instructions. The RNA quality was assessed on a NanoDrop 2000 (Thermo), where the 260/280 ratio was obtained. Samples with a ratio of 1.8–2.0 were processed for downstream gene analysis. 1,000 ng of RNA was reverse-transcribed into cDNA using a PrimeScript RT Reagent Kit

with gDNA Eraser (Takara) according to the manufacturer's protocol. qRT–PCR was performed with SYBR Premix Ex Taq II (Takara) on a Bio-Applied Biosystems 7300 (Life Technology). The results were analyzed using the comparative cycle threshold (CT) method. The primers are presented in Appendix Table S2.

**mGPDH activity assay**

The mGPDH activity was measured using mitochondria, which were separated from total homogenate according to the manufacturer's recommendation of the Mitochondria Isolation Kit (Beyotime). The activity was detected using 2,6-dichloroindophenol (DCIP; Sigma) as the electron acceptor and measuring the loss of absorbance at 600 nm (reaction buffer consisted of 50 mM of $KH_2PO_4/K_2HPO_4$ buffer (pH 7.5), 9.3 μM of antimycin A, 5 μM of rotenone, and 50 μM of DCIP; reaction conducted at 37°C).

**Immunofluorescence staining**

Cells and tissue sections were fixed in 4% paraformaldehyde for 30 min, permeabilized with 0.1–0.2% Triton X-100 (Sigma) at room temperature (RT) for 10–15 min, blocked with 5% BSA for 30 min at RT, and incubated with the indicated primary antibodies (1:50–1:100) overnight at 4°C, followed by their corresponding secondary antibodies for 1 h at RT. Nuclei were stained with DAPI (Beyotime) for 5 min, and the coverslips were mounted with Fluoromount (BOSTER); images were acquired using a fluorescence microscope (Olympus cellSens standard 1.15). The fusion index was calculated as the ratio of the number of nuclei incorporated into myotubes (> 2 nuclei) to the total number of nuclei. Nuclei were counted from five images/dish using ImageJ. Nuclear number assays were determined by calculating the percentage of nuclei incorporated into the MyHC-positive myotubes with the indicated number of nuclei.

**Human clinical studies**

All experimental protocols were approved by the Ethics Committee of Xinqiao Hospital, Third Military Medical University, and were registered online (Clinical trial register no. ChiCTR-ROC-17010719), and conformed to the principles set out in the WMA Declaration of Helsinki and the Department of Health and Human Services Belmont Report. All participants provided written informed consent. A total of 11 obese patients and 18 normal subjects participated in this study. Obesity was defined according to the WHO Western Pacific Regional Office definition (Wen *et al*, 2009). Biopsies of GA muscles were obtained from all participants, and the clinical characteristics are presented in Appendix Table S1.

**Statistical analyses**

All data were analyzed using GraphPad Prism 7 (Macintosh). Quantitative values are presented as the mean ± s.e.m. Statistical differences between two experimental groups were analyzed using two-tailed Student's *t*-test. The mean IHC scores were analyzed using the Wilcoxon test. The distribution of nuclei per myotube and the fiber cross-sectional area was analyzed using the Kolmogorov–Smirnov test. *P*-values < 0.05 were considered significant. Sample sizes were estimated on the basis of our previous experiences

(Zheng *et al*, 2011; Wang *et al*, 2016), and no samples were excluded from the study. All *P*-values for the main figures, expanded view figures, and appendix figures are presented in Dataset EV1.

**Expanded View** for this article is available online.

## Acknowledgements

This work was supported by grants from The National Natural Science Foundation of China (No. 81700714, No. 81721001, No. 81471039, No. 81270893, No. 81401601, and No. 81600673), the National Key R&D Program of China (No. 2016YFC1101100 and No. 2017YFC1309602), and the Natural Science Foundation Project of Chongqing (No. cstc2014jcyjjq10006, No. cstc2016jcyjA0518, No. cstc2016jcyjA0093, and No. cstc2017jcyjA1192).

## Author contributions

HQ, XL, YZ, QL, and LZ: acquisition of data, analysis and interpretation of data, and statistical analysis; HQ, YZ, and QL: drafting of the manuscript; HQ, XL, XX, YW, RZ, QT, HW, and JX: analysis and interpretation of data; ZZ, GY, ZL, and HD: critical revision of the manuscript for important intellectual content; HZ: study concept and design, analysis and interpretation of data, drafting of the manuscript, critical revision of the manuscript for important intellectual content, obtaining study funding, and study supervision.

## Conflict of interest

The authors declare that they have no conflict of interest.

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
