## [Review Process File · EMBO Molecular Medicine]

Mitochondrial glycerol 3-phosphate dehydrogenase promotes skeletal muscle regeneration

Xiufei Liu, Hua Qu, Yi Zheng, Qian Liao¹, Linlin Zhang, Xiaoyu Liao, Xin Xiong, Yuren Wang, Rui Zhang, Hui Wang, Qiang Tong, Zhenqi Liu, Hui Dong, Gangyi Yang, Zhiming Zhu, Jing Xu, and Hongting Zheng

Review timeline:

Submission date:	31 May 2018
Editorial Decision:	03 July 2018
Revision received:	14 September 2018
Editorial Decision:	01 October 2018
Revision received:	10 October 2018
Accepted:	11 October 2018

Editor: Céline Carret

Transaction Report:

1st Editorial Decision

03 July 2018

Thank you for the submission of your manuscript to EMBO Molecular Medicine. We have now heard back from the two referees whom we asked to evaluate your manuscript. Although the referees find the study to be of potential interest, they also raise a number of concerns that need to be fully addressed in a major revision of your article.

You will see from the set of comments pasted below that Referee 1 would like to see more mechanism and offers suggestions to achieve that. Referee 2 requests better stats, imaging and illustrations, western blots, a more detailed method and clarifications, all being needed to make the data more meaningful. Both referees ask for a careful check of typos and grammar.

Upon our cross-commenting exercise, referees agreed that in terms of molecular mechanism, what would be really needed is the evaluation of siRNA cGPDH in vitro and most importantly, the exploration of the NAD/NADPH ratio, and then if possible, the PGC1- α acetylation level, even though Referee 1 agrees that this is a tricky experiment that can be difficult to achieve.

We would welcome the submission of a revised version within three months for further consideration and would like to encourage you to address all the criticisms raised as suggested to improve conclusiveness and clarity. Please note that EMBO Molecular Medicine strongly supports a single round of revision and that, as acceptance or rejection of the manuscript will depend on another round of review, your responses should be as complete as possible.

Please also contact us as soon as possible if similar work is published elsewhere. If other work is published we may not be able to extend the revision period beyond three months. Please read below for important editorial formatting and consult our author's guidelines for proper formatting of your revised article for EMBO Molecular Medicine.

I look forward to receiving your revised manuscript.

***** Reviewer's comments *****

Referee #1 (Comments on Novelty/Model System for Author):

The studies were very well done with proper control and statistical information. The data are generally robust.

Referee #1 (Remarks for Author):

The regenerative function of adult skeletal muscle is critical for maintaining tissue homeostasis. In this study, the authors described a new pathway involving the mitochondrial mGPDH that appears to have potent effects on muscle regeneration. mGPDH expression is highly induced during C2C12 myotube differentiation and its levels are linked to the activation of mature myotube markers. The authors demonstrated that mGPDH deficiency impaired muscle regeneration following cardiotoxin injection while its overexpression improved muscle histology and function in Mdx mice. The significance of mGPDH in muscle regeneration is further assessed using the CTX model in the context of diet-induced obesity and STZ-induced diabetes. The results support a crucial function of mGPDH in driving muscle regeneration. Finally, the authors provided evidence to implicate CaMKK β -AMPK in mediating the effects of mGPDH on mitochondrial biogenesis.

Overall, this study provided compelling evidence to support mGPDH as an important regulator of the regenerative function of skeletal muscle. The *in vivo* data are particularly strong. The manuscript is well written. The manuscript can be further strengthened with additional studies to probe the mechanisms in greater detail. Several points should be addressed.

1. As an important aspect of this study is muscle regeneration in obesity and diabetes, the authors should provide evidence that the regenerative function of muscle is impaired in HFD-fed mice and in STZ-injected mice. This would support the observed reduction of skeletal muscle mGPDH expression in these models.
2. A major function of mGPDH is to form the glycerol-3-phosphate shuttle with cytosolic GPDH. Does RNAi knockdown of cytosolic GPDH also impair C2C12 differentiation? Is expression of cytosolic GPDH altered in obesity, diabetes and CTX/MDX models? These analyses would help clarify whether the glycerol-3-phosphate shuttle per se is important for regeneration.
3. Does mGPDH gain and loss of function manipulations affect NAD/NADH ratio? If so, it would be important to explore whether altered NAD levels and PGC-1 α acetylation may account for the link between mGPDH and mitochondrial biogenesis. At least this should be explored using transiently transfection of PGC-1 α and mGPDH.
4. Inflammatory signaling is an important player in the regulation of tissue repair and regeneration. Additional data on cytokine gene expression and signaling should be included for the *in vivo* studies.

Referee #2 (Comments on Novelty/Model System for Author):

- few figures should be redone for clarity with appropriate statistical analysis (see below)
- solid data
- stimulating mitochondrial biogenesis through mGDPH (or any other strategy) won't cure DMD but could be a potential complementary strategy to gene therapies.

Referee #2 (Remarks for Author):

The purpose of this paper is to determine the role of the mGPDH in the regulation the myogenesis through CaMKKb/AMPK pathway by stimulating the mitochondrial biogenesis. These corroborate with previous papers published on this matter, describing the consequences of mitochondria depletion in C2C12 cells on their differentiation capacity, following the mitochondrial biogenesis during the muscle regeneration and the impact of mitochondrial metabolism on myoblasts differentiation. Some of these papers could be added in their discussion. The authors then explore the impact of the regulation of mGPDH expression level on the skeletal muscle regeneration/mass in different conditions such as obesity and muscular dystrophy (DMD).

This paper contains an impressive amount of solid data. There are only minor elements that can be improve to increase the quality of the data.

Major points:

- In their introduction and discussion, the authors should moderate their words. When they say that they will improve the "impaired muscle regeneration" in dmd mice. These mice don't have a low capacity to regenerate their muscles, in contrary as suggested by the increased number of central myonuclei over age (Duddy et al, Skeletal muscle, 2015). These mice, because of dystrophin deficiency, keep loosing/damaging their fragile myofibers, but can still regenerate their fibers.
- Figure 1: Statistic are missing in the figures 1a, 1b, 1c.
- Figure 1g: Kolmogorov Smirnov test to show the differences in the distribution of nuclei per myotubes would be better than t-test.
- Supplemental figure 2: it seems that the expression level of mGPDH vary muscles to muscles, and does not match with the muscle fiber type. Can the authors confirmed that it is indeed the case: mGPDH does not correlate with fiber types I, IIa or IIb?
- Figure 2a and c: it seems that the mGDPH level is increased only in the first few days of regeneration. Does this level correlate with developmental myosin heavy chain? Is mGPDH present mainly in newly formed myofibres in vivo?
- Figure 2d: the authors should give the percentage of myofibers with central nuclei, and show a distribution of the fibres CSA.
- Supplemental 2g: the authors should show a distribution of fibre size (CSA) instead of a global mean.
- Figure 2g: desmin immunostaining is not convincing.
- Figure 2i: the authors could quantify the necrotic area
- Supplemental 3: An immunostaining of Pax7 and counting the number of satellite cells per fibres would be more accurate than an RTqPCR on total muscle extract (the level of Pax7 mRNA could be underestimated, as the amount of satellites cells is low compare to muscle fibres). Idem for MyoD.
- Figure 2g: which muscle was weighted? It is important to know, as the expression level of mGPDH varies muscle to muscle according to their Supplemental figure 2a.
- Figure 2i: does not match with collagen deposit and trichrome staining. I guess the author mean Fig 2 q in the text instead (line 137, page 5). The collagen area should be still quantified.
- Page 5 line 149: "insufficient muscle regeneration" is not correct for DMD: the mice regenerate very well their muscles, but due to their muscle fragility because of dystrophin deficient myofiber keep regenerating.
- Supplementary figure 4: what age were the mdx mice, which muscle has been analysed (as mGDPH level varies muscle to muscle), number of mice analysed, western blot should be quantified.
- Figure 2n: Kolmogorov smirnov test should be used to compare the distribution of the mdx-GFP and mdx-mGPDH mice CSA.
- Figure2p: it is not a proper utrophin staining. The immunostaining should be sharper at the periphery of the myofibers and not localized in the interspace between myofibers.
- Figure 2r: the authors should give the efficiency of AAV to target different muscles and what is the level of mGPDH?
- Figure3k: the authors should give the fusion index and the number of myonuclei per myotubes in the different conditions.
- Figure4g and k: a distribution of the CSA would be a better representation of the data, rather than a global mean.
- Western blot: there are some inconsistency in the band for mGDPH western to western: sometimes the band is net and sharp, and sometimes it appears more blur/wider.

Minor points:

- References formatting should be redone.

Referee #1 (Comments on Novelty/Model System for Author):

The studies were very well done with proper control and statistical information. The data are generally robust.

(Remarks for Author)

The regenerative function of adult skeletal muscle is critical for maintaining tissue homeostasis. In this study, the authors described a new pathway involving the mitochondrial mGPDH that appears to have potent effects on muscle regeneration. mGPDH expression is highly induced during C2C12 myotube differentiation and its levels are linked to the activation of mature myotube markers. The authors demonstrated that mGPDH deficiency impaired muscle regeneration following cardiotoxin injection while its overexpression improved muscle histology and function in Mdx mice. The significance of mGPDH in muscle regeneration is further assessed using the CTX model in the context of diet-induced obesity and STZ-induced diabetes. The results support a crucial function of mGPDH in driving muscle regeneration. Finally, the authors provided evidence to implicate CaMKK β -AMPK in mediating the effects of mGPDH on mitochondrial biogenesis.

Overall, this study provided compelling evidence to support mGPDH as an important regulator of the regenerative function of skeletal muscle. The *in vivo* data are particularly strong. The manuscript is well written. The manuscript can be further strengthened with additional studies to probe the mechanisms in greater detail. Several points should be addressed.

1. As an important aspect of this study is muscle regeneration in obesity and diabetes, the authors should provide evidence that the regenerative function of muscle is impaired in HFD-fed mice and in STZ-injected mice. This would support the observed reduction of skeletal muscle mGPDH expression in these models.

A: We performed additional experiments per the reviewer's suggestions. In HFD-fed mice (HFD mice were maintained in our lab for other projects) and STZ-injected mice, the regenerative function of muscle was impaired, which manifested as the decreased differentiation markers myogenin and myh3, with a delay in the disappearance of necrotic fibers and fewer and more unevenly distributed newly formed myofibers with multiple centrally located nuclei (Figure 1A-H). These results have been included in our revised version as Fig EV3.

Figure 1. Skeletal muscle regeneration is impaired in HFD-fed mice and STZ-injected mice.

A-D Gastrocnemius (GA) muscles were obtained from HFD-fed mice at day 7 post-CTX injury. Quantification of myogenin and myh3 by qRT-PCR (A), representative images of the H&E staining (B), distribution of the fiber cross-section area (CSA) (C), and percentage of myofibers with central nuclei (D).

E-H GA muscles were obtained from STZ-treated mice 4 weeks after STZ injection and at day 7 post-CTX injury. Quantification of myogenin and myh3 by qRT-PCR (E), representative images of the H&E staining (F), distribution of the fiber CSA (G), and percentage of myofibers with central nuclei (H).

Data information: Data are presented as the mean \pm s.e.m. Scale bars represent 100 μ m in panels B and F. $n = 6$ mice per group. * $P < 0.05$, ** $P < 0.01$, *** $P < 0.001$. Unpaired t -test was used for all analyses except in panel C and G, where Kolmogorov-Smirnov test was used.

2. A major function of mGPDH is to form the glycerol-3-phosphate shuttle with cytosolic GPDH. Does RNAi knockdown of cytosolic GPDH also impair C2C12 differentiation? Is expression of cytosolic GPDH altered in obesity, diabetes and CTX/MDX models? These analyses would help clarify whether the glycerol-3-phosphate shuttle per se is important for regeneration.

A: We supplemented observations related to cytosolic GPDH (cGPDH) according to your suggestions. During the differentiation process of C2C12 myocytes, the expression of cGPDH had no significant change, and the knockdown of cGPDH by corresponding siRNA showed no significant effects on C2C12 differentiation (Figure 2A-F). Moreover, cGPDH expressions were not notably altered in the obesity, diabetes and CTX/MDX models (Figure 2G). We are grateful for these important suggestions for vital mechanism clarification, and these results have been included in our revised version as Fig EV1 and Appendix Fig S9.

Figure 2. The effect of cGPDH on myoblast differentiation and skeletal muscle regeneration.
A cGPDH expression during C2C12 myocyte differentiation.

B-D Representative images of MyHC immunofluorescence (B) of C2C12 myocytes transfected with siRNA targeting cGPD; the fusion index (C) and the distribution of nuclei per myotube (D) were calculated.

E, F qRT-PCR (E) and western blot analysis (F) of myogenin and MyHC in C2C12 myocytes transfected with siRNA targeting cGPDH.

G Western blot analysis of cGPDH in GA muscles of indicated mouse models.

Data information: Data are presented as the mean \pm s.e.m. Scale bars represent 50 μ m in panel B. In panels A-F, $n = 3$; in panel G, $n = 3$ mice per group. $*P < 0.05$. Unpaired t -test was used for all analyses except in panel D, where the Kolmogorov-Smirnov test was used.

3. Does mGPDH gain and loss of function manipulations affect NAD/NADH ratio? If so, it would be important to explore whether altered NAD levels and PGC-1 α acetylation may account for the link between mGPDH and mitochondrial biogenesis. At least this should be explored using transiently transfection of PGC-1 α and mGPDH.

A: We conducted new experiments to measure the NAD⁺/NADH ratio and PGC-1 α acetylation according to your suggestion. mGPDH affected the ratio of NAD⁺/NADH (Figure 3A). Moreover, PGC-1 α acetylation was altered when transiently transfected PGC-1 α and mGPDH (Figure 3B). These results and relevant references (Iwabu et al, 2010; Meng et al, 2013; Woldt et al, 2013) have been included in our revised version of the manuscript as Fig 3J and K. Thank you for this helpful comment, which led us to strengthen the mechanisms of mGPDH in greater detail.

Figure 3. mGPDH regulates NAD⁺/NADH ratio and PGC-1 α acetylation.

A NAD⁺/NADH ratio was assessed in C2C12 myocytes transfected with siRNA or plasmid for mGPDH.

B Acetyl-lysine (Ac-Lys) level of PGC-1 α was analyzed by immunoprecipitation.

Data information: Data are presented as the mean \pm s.e.m. $n = 3$. $*P < 0.05$, $**P < 0.01$. Unpaired t -test was used.

4. Inflammatory signalling is an important player in the regulation of tissue repair and regeneration. Additional data on cytokine gene expression and signalling should be included for the in vivo studies.

A: We have included new observations on inflammatory cytokines in our studies according to your suggestion. Inflammatory signalling is an important player in the regulation of tissue repair and regeneration. Following injury, the inflammatory response ensues, which promotes the removal of necrotic tissue and repairs damaged skeletal muscle tissue (De Bleecker & Engel, 1994). Our results showed that during the muscle regeneration process of HFD and STZ mice, inflammatory cytokine gene expressions were reduced (Figure 4A), which were consistent with previous studies (Brown et al, 2015; Nguyen et al, 2011), indicating an impaired muscle regeneration process. However, although mGPDH ablation (mGPDH^{-/-} mice) or overexpression (AAV-mGPDH) showed relatively decreased or increased trends of cytokine genes, respectively, their expressions were not statistically significant with the exception of IL-1 β in mGPDH^{-/-} mice (Figure 4B,C). These data have been included in our revised version as Fig EV5.

Figure 4. The effect of mGPDH on inflammatory signaling.

A-C Quantification of indicated inflammatory cytokines by qRT-PCR in GA muscles of HFD-fed and STZ-treated mice (A), mGPDH^{-/-} mice (B), and HFD-fed and STZ-treated mice intramuscularly injected with AAV-mGPDH (C) at day 7 post-CTX injury.

Data information: Data are presented as the mean \pm s.e.m. $n = 6$ mice per group. * $P < 0.05$, ** $P < 0.01$. Unpaired t -test was used for all panels.

Referee #2

(Comments on Novelty/Model System for Author)

- few figures should be redone for clarity with appropriate statistical analysis (see below)
- solid data
- stimulating mitochondrial biogenesis through mGPDH (or any other strategy) won't cure DMD but could be a potential complementary strategy to gene therapies.

(Remarks for Author)

The purpose of this paper is to determine the role of the mGPDH in the regulation the myogenesis through CaMKKb/AMPK pathway by stimulating the mitochondrial biogenesis. These corroborate with previous papers published on this matter, describing the consequences of mitochondria depletion in C2C12 cells on their differentiation capacity, following the mitochondrial biogenesis during the muscle regeneration and the impact of mitochondrial metabolism on myoblasts differentiation. Some of these papers could be added in their discussion. The authors then explore the impact of the regulation of mGPDH expression level on the skeletal muscle regeneration/mass in different conditions such as obesity and muscular dystrophy (DMD).

This paper contains an impressive amount of solid data. There are only minor elements that can be improve to increase the quality of the data.

A: Previous studies have indicated the effect of mitochondrial depletion on myoblast differentiation. The corresponding content and references (Cerletti et al, 2012; Duguez et al, 2012; Varaljai et al, 2015) have been included in our revised discussion.

Major points:

1- In their introduction and discussion, the authors should moderate their words. When they say that they will improve the "impaired muscle regeneration" in dmd mice. These mice don't have a low capacity to regenerate their muscles, in contrary as suggested by the increased number of central myonuclei over age (Duddy et al, Skeletal muscle, 2015). These mice, because of dystrophin deficiency, keep losing/damaging their fragile myofibers, but can still regenerate their fibers.

A: We appreciate this reviewer's comment. It is the continuous loss of damaged myofibers, rather than impaired muscle regeneration, that leads to muscle pathologies in dmd mice. We have changed the description of "impaired muscle regeneration" in dmd mice in the introduction and discussion sections and have cited relevant references (Barton et al, 2002; Duddy et al, 2015; Novak et al, 2017). Moreover, we recognize that mGPDH would not cure DMD; however, it may be a potential complementary strategy to gene therapies. Thus, these statements have been modified in our revised version.

2- Figure 1: Statistic are missing in the figures 1a, 1b, 1c.

A: We have supplemented these missed statistics.

3- Figure 1g: Kolmogorov Smirnov test to show the differences in the distribution of nuclei per myotubes would be better than t-test.

A: The statistics of Figure 1g have been changed to the Kolmogorov-Smirnov test.

4- Supplemental figure 2: it seems that the expression level of mGPDH vary muscles to muscles, and does not match with the muscle fiber type. Can the authors confirmed that it is indeed the case: mGPDH does not correlate with fiber types I, IIa or IIb?

A: In Supplemental Figure 2, the expression level of mGPDH varies among muscles, and it seems that mGPDH does not match with the muscle fiber type. We have conducted additional experiments to observe this issue. GA is type I and type II mixed muscle; however, it is mainly composed of type IIb. Thus, we co-stained MHC IIb (red) with mGPDH (green). As shown in Figure 5A, fibers were stained yellow (mGPDH and MHC IIb co-staining), green (mGPDH separate staining) and red (MHC IIb separate staining), indicating mGPDH did not match with the fiber type in GA muscle. Furthermore, we assessed whether the muscle fiber type could be regulated by mGPDH, and the results showed the expression of MHC isoforms (MHC I, IIa and IIb) were not significantly changed in the mGPDH-depleted skeletal muscle (Figure 5B). These results have been included in our revised version as Fig EV2B and C.

Figure 5. The relationship between mGPDH with muscle fiber types.

A Immunofluorescence showing localization of mGPDH with fiber type marker MHC-IIb on cryosections from uninjured GA muscle of 8-week-old C57BL/6J mice.

B qRT-PCR analyses of indicated fiber type markers (MHC I, IIa and IIb) in the uninjured GA muscles of 8-week-old WT and mGPDH^{-/-} mice.

Data information: Data are presented as the mean \pm s.e.m. Scale bars represent 200 μ m in panel A. $n = 6$ mice per group. Unpaired t -test was used for all panels.

5- Figure 2a and c: it seems that the mGDPH level is increased only in the first few days of regeneration. Does this level correlate with developmental myosin heavy chain? Is mGPDH present mainly in newly formed myofibres in vivo?

A: To further assess whether mGPDH correlates with developmental myosin heavy chain (dMHC), we supplemented the observations of dMHC expression, including the myosin heavy chain 3 (myh3), myh8 and myosin light chain 4 (myl4) after CTX injury (Schiaffino et al, 2015). These dMHCs shared the same expression pattern, i.e., significantly upregulated at day 3 and peaked at day 7 after CTX injury (Figure 6A), which is consistent with the pattern of myogenin (Figure 2A in original version). Although the mRNA expression of mGPDH peaked at day 3, it was maintained a high level to day 7 (Figure 2A in original version) and has a certain overlap with the increase of dMHC. This earlier but overlaid expression pattern indicated that the mGPDH was correlated with or drove the expressions of dMHC. The regulatory role of mGPDH on dMHC could be further confirmed by mGPDH gain or loss of function, which indicated the myh3 expression was decreased after mGPDH knockout and increased after mGPDH overexpression (Figure 2 J and I in original version). As dMHC was mainly expressed in newly generated fiber and mGPDH correlated with dMHC, mGPDH was likely to present mainly in newly formed myofibers. To confirm this, we stained mGPDH in GA muscle post CTX intramuscular injection. The results showed that compared with the basal expression of mGPDH in normal fibers with peripheral nuclei, the injury-induced higher expression of mGPDH was mainly localized in regenerating fibers with central nuclei (Figure 6B), which indicates the injury-induced mGPDH expression presented mainly in newly formed myofibers in vivo. These findings have been included in our revised version as Fig 2A and Appendix Fig S2, and thank you for the suggestion.

Figure 6. The relationship between mGPDH with developmental myosin heavy chain.

A qRT-PCR analyses of indicated developmental myosin heavy chain (dmHC, myh8, myl4 and myh3) in the GA muscles from C57BL/6J mice at day 7 post CTX intramuscular injection. B Immunostaining of mGPDH in the GA muscles from C57BL/6J mice at day 7 post CTX intramuscular injection.

Data information: Data are presented as the mean \pm s.e.m. Scale bars represent 100 μ m in panel B. $n = 3$ mice per group. * $P < 0.05$, *** $P < 0.001$. Unpaired t -test was used for all panels.

6- Figure 2d: the authors should give the percentage of myofibers with central nuclei, and show a distribution of the fibres CSA.

A: According to your suggestions, we have switched the representation in Figure 2d to the percentage of myofibers with central nuclei and have included a distribution of the fiber CSA.

7- Supplemental 2g: the authors should show a distribution of fibre size (CSA) instead of a global mean.

A: We have changed to a distribution of fiber CSA in Supplemental 2g (original version) and updated as Fig EV2I in our revised version.

8- Figure 2g: desmin immunostaining is not convincing.

A: Desmin immunostaining was re-conducted and is shown as follows; the old image of Figure 2g has been replaced.

Figure 7. Immunostaining for desmin.

Immunofluorescence staining of desmin (red) in GA muscle from WT and mGPDH^{-/-} mice at day 7 post CTX injury.

Data information: Scale bars represent 50 μ m. $n = 6$ mice per group.

9- Figure 2i: the authors could quantify the necrotic area

A: Quantification has been performed for Figure 2i according to your suggestion.

10- Supplemental 3: An immunostaining of Pax7 and counting the number of satellite cells per fibres would be more accurate than an RTqPCR on total muscle extract (the level of Pax7 mRNA could be underestimated, as the amount of satellites cells is low compare to muscle fibres). Idem for MyoD.

A: According to your suggestions, we performed immunostaining of Pax7 and MyoD and counted the number of satellite cells per fiber. The data were incorporated in the revised version as Appendix Fig S3.

Figure 8. Immunostaining for PAX7 and MyoD.

A-D Immunofluorescence staining of PAX7 (green, A) and MyoD (green, C) and their corresponding quantifications (B and D) in GA muscles from WT and mGPDH^{-/-} mice at day 7 post CTX injury.

Data information: Scale bars represent 50 μ m in panels A and C. $n = 6$ mice per group.

11- Figure 2g: which muscle was weighted? It is important to know, as the expression level of mGPDH varies muscle to muscle according to their Supplemental figure 2a.

A: GA muscle was weighted in this Figure, and we have included these data in the corresponding results section and legend of Figure 2g, which has been updated as Fig 2H in our revised version.

12- Figure 2i: does not match with collagen deposit and trichrome staining. I guess the author mean Fig 2 q in the text instead (line 137, page 5). The collagen area should be still quantified.

A: We have revised these results and legends accordingly, and the collagen areas have been quantified.

13- Page 5 line 149: "insufficient muscle regeneration" is not correct for DMD: the mice regenerate very well their muscles, but due to their muscle fragility because of dystrophin deficient myofiber keep regenerating.

A: We appreciated your kind reminder. We have changed the incorrect description "insufficient muscle regeneration" for DMD and have cited the relevant references.

14- Supplementary figure 4: what age were the mdx mice, which muscle has been analysed (as mGDPH level varies muscle to muscle), number of mice analysed, western blot should be quantified.

A: In supplementary figure 4, the age of the mdx mice is 12 w, the GA muscle has been analyzed, and $n = 3$ mice per group. This information has been included in the corresponding result and legend sections, and the western blot has been quantified.

15- Figure 2n: Kolmogorov smirnov test should be used to compare the distribution of the mdx-GFP and mdx-mGDPH mice CSA.

A: We have changed the statistical method to the Kolmogorov-Smirnov test in Figure 2n.

16- Figure 2p: it is not a proper utrophin staining. The immunostaining should be sharper at the periphery of the myofibers and not localized in the interspace between myofibers.

A: We re-conducted utrophin immunostaining to replace the previous image.

Figure 9. Immunostaining for utrophin.

Immunofluorescence staining of utrophin (red) in GA muscle from mdx mice 4 weeks after AAV-mGDPH intramuscular injection.

Data information: Scale bars represent 50 μm . $n = 6$ mice per group.

17- Figure 2r: the authors should give the efficiency of AAV to target different muscles and what is the level of mGDPH?

A: The efficiency of AAV-mGPDH to target different muscles has been presented as follows. Thank you for your kind reminder, and this information has been included in our revised version as Appendix Fig S5.

Figure 10. The efficiency of AAV-mGPDH to target different muscles.

Mdx mice were treated with AAV-mGPDH via tail vein, and after 4 weeks, the mGPDH levels in indicated muscles were analyzed by qRT-PCR.

Data information: Data are presented as the mean \pm s.e.m. * $P < 0.05$, ** $P < 0.01$. Unpaired t -test was used for all panels.

18- Figure3k: the authors should give the fusion index and the number of myonuclei per myotubes in the different conditions.

A: We have supplemented the fusion index and the number of myonuclei per myotubes for Figure 3k, which has been updated as Fig 3O and P in our revised version.

19- Figure4g and k: a distribution of the CSA would be a better representation of the data, rather than a global mean.

A: We have changed to a distribution of fiber CSA in Figure 4g and k.

20- Western blot: there are some inconsistency in the band for mGDPH western to western: sometimes the band is net and sharp, and sometimes it appears more blur/wider.

A: There were two commercial antibodies (listed in “western blot and antibodies” of Materials and methods) for detecting mGPDH protein expression in our experiments. Both antibodies shared the same band position and were proven correct by previous studies (Madiraju et al, 2014; Thakur et al, 2018). As the mGPDH antibody from Santa-Cruz Biotechnology has relatively low sensitivity for in vivo experiments, we adopted the Abcam mGPDH antibody to conduct supplemental experiments. We appreciate your kind reminder, and we will attempt to use the same antibody in our future studies.

Figure 11. Comparison of antibodies against mGPDH from different commercial companies. mGPDH protein expressions were detected in GA muscles from C57BL/6J mice using antibodies against mGPDH from Santa-Cruz Biotechnology and Abcam, respectively. m: marker.

Minor points:

21- References formatting should be redone.

A: Reference formatting has been redone.

References

- Barton ER, Morris L, Musaro A, Rosenthal N, Sweeney HL (2002) Muscle-specific expression of insulin-like growth factor I counters muscle decline in mdx mice. *J Cell Biol* 157: 137-148
- Brown LA, Lee DE, Patton JF, Perry RA, Jr., Brown JL, Baum JI, Smith-Blair N, Greene NP, Washington TA (2015) Diet-induced obesity alters anabolic signalling in mice at the onset of skeletal muscle regeneration. *Acta Physiol (Oxf)* 215: 46-57
- Cerletti M, Jang YC, Finley LW, Haigis MC, Wagers AJ (2012) Short-term calorie restriction enhances skeletal muscle stem cell function. *Cell Stem Cell* 10: 515-519
- De Bleeker JL, Engel AG (1994) Expression of cell adhesion molecules in inflammatory myopathies and Duchenne dystrophy. *J Neuropathol Exp Neurol* 53: 369-376
- Duddy W, Duguez S, Johnston H, Cohen TV, Phadke A, Gordish-Dressman H, Nagaraju K, Gnocchi V, Low S, Partridge T (2015) Muscular dystrophy in the mdx mouse is a severe myopathy compounded by hypotrophy, hypertrophy and hyperplasia. *Skeletal muscle* 5: 16
- Duguez S, Duddy WJ, Gnocchi V, Bowe J, Dadgar S, Partridge TA (2012) Atmospheric oxygen tension slows myoblast proliferation via mitochondrial activation. *PLoS One* 7: e43853
- Iwabu M, Yamauchi T, Okada-Iwabu M, Sato K, Nakagawa T, Funata M, Yamaguchi M, Namiki S, Nakayama R, Tabata M, Ogata H, Kubota N, Takamoto I, Hayashi YK, Yamauchi N, Waki H, Fukayama M, Nishino I, Tokuyama K, Ueki K et al (2010) Adiponectin and AdipoR1 regulate PGC-1alpha and mitochondria by Ca(2+) and AMPK/SIRT1. *Nature* 464: 1313-1319
- Madiraju AK, Erion DM, Rahimi Y, Zhang XM, Braddock DT, Albright RA, Prigaro BJ, Wood JL, Bhanot S, MacDonald MJ, Jurczak MJ, Camporez JP, Lee HY, Cline GW, Samuel VT, Kibbey RG, Shulman GI (2014) Metformin suppresses gluconeogenesis by inhibiting mitochondrial glycerophosphate dehydrogenase. *Nature* 510: 542-546
- Meng ZX, Li S, Wang L, Ko HJ, Lee Y, Jung DY, Okutsu M, Yan Z, Kim JK, Lin JD (2013) Baf60c drives glycolytic metabolism in the muscle and improves systemic glucose homeostasis through Deptor-mediated Akt activation. *Nature medicine* 19: 640-645
- Nguyen MH, Cheng M, Koh TJ (2011) Impaired muscle regeneration in ob/ob and db/db mice. *ScientificWorldJournal* 11: 1525-1535
- Novak JS, Hogarth MW, Boehler JF, Nearing M, Vila MC, Heredia R, Fiorillo AA, Zhang A, Hathout Y, Hoffman EP, Jaiswal JK, Nagaraju K, Cirak S, Partridge TA (2017) Myoblasts and macrophages are required for therapeutic morpholino antisense oligonucleotide delivery to dystrophic muscle. *Nature communications* 8: 941
- Schiaffino S, Rossi AC, Smerdu V, Leinwand LA, Reggiani C (2015) Developmental myosins: expression patterns and functional significance. *Skelet Muscle* 5: 22
- Thakur S, Daley B, Gaskins K, Vasko VV, Boufraqueh M, Patel D, Sourbier C, Reece J, Cheng SY, Kebebew E, Agarwal S, Klubo-Gwiedzinska J (2018) Metformin Targets Mitochondrial Glycerophosphate Dehydrogenase to Control Rate of Oxidative Phosphorylation and Growth of Thyroid Cancer In Vitro and In Vivo. *Clin Cancer Res* 24: 4030-4043
- Varaljai R, Islam AB, Beshiri ML, Rehman J, Lopez-Bigas N, Benevolenskaya EV (2015) Increased mitochondrial function downstream from KDM5A histone demethylase rescues differentiation in pRB-deficient cells. *Genes Dev* 29: 1817-1834
- Woldt E, Sebti Y, Solt LA, Duhem C, Lancel S, Eeckhoutte J, Hesselink MK, Paquet C, Delhaye S, Shin Y, Kamenecka TM, Schaart G, Lefebvre P, Neviere R, Burris TP, Schrauwen P, Staels B, Duez H (2013) Rev-erb-alpha modulates skeletal muscle oxidative capacity by regulating mitochondrial biogenesis and autophagy. *Nat Med* 19: 1039-1046

2nd Editorial Decision

01 October 2018

Thank you for the submission of your revised manuscript to EMBO Molecular Medicine. We have now received the enclosed reports from the referees that were asked to re-assess it. As you will see the reviewers are now globally supportive and I am pleased to inform you that we will be able to accept your manuscript pending minor editorial amendments as well as a response to the referee's final comments.

Please submit your revised manuscript within two weeks. I look forward to seeing a revised form of your manuscript as soon as possible.

***** Reviewer's comments *****

Referee #1 (Remarks for Author):

The authors provided new data to support the conclusions and have adequately addressed my concerns. Some minor changes on grammar and wording would be a further plus for this manuscript.

Referee #2 (Remarks for Author):

The authors responded to all my comments.
I would have two more remarks for them:

Paragraph: mGPDH is essential to skeletal muscle regeneration

The authors should change "Myoblast differentiation occurs in two distinct phases" to "Myoblast differentiation occurs during muscle development and also during adulthood for muscle mass maintenance and muscle regeneration (Charge & Rudnicki, 2004). Here, we aim to identify the role of mGPDH in both stages.

" as the word "phases" is confusing in this context.

Line 254-255: The authors should rephrase this sentence:

"While cGPDH expressions were not altered notably in obesity, diabetes and CTX/mdx models (Appendix Fig S9)."

2nd Revision - authors' response

10 October 2018

***** Reviewer's comments *****

Referee #1 (Remarks for Author):

The authors provided new data to support the conclusions and have adequately addressed my concerns. Some minor changes on grammar and wording would be a further plus for this manuscript.

A: Thank you for your suggestion. We have employed a professional editing company to improve the language and correct the grammatical errors.

Referee #2 (Remarks for Author):

The authors responded to all my comments.
I would have two more remarks for them:

Paragraph: mGPDH is essential to skeletal muscle regeneration

The authors should change "Myoblast differentiation occurs in two distinct phases" to "Myoblast differentiation occurs during muscle development and also during adulthood for muscle mass maintenance and muscle regeneration (Charge & Rudnicki, 2004). Here, we aim to identify the role of mGPDH in both stages. " as the word "phases" is confusing in this context.

A: We appreciate your helpful suggestion. We have changed this sentence accordingly.

Line 254-255: The authors should rephrase this sentence: "While cGPDH expressions were not altered notably in obesity, diabetes and CTX/mdx models (Appendix Fig S9)."

A: We have rephrased this sentence as follows: "In contrast, the cGPDH expressions were not notably altered in obese (HFD), diabetic (STZ) and mdx mice. Moreover, the expressions did not change during the regeneration process post CTX injection (Appendix Fig S9)".

Corresponding Author Name: Hongting Zheng

Manuscript Number: EMM-2018-09390